# Flowering Biology of Selected Hybrid Grape Cultivars under Temperate Climate Conditions

**Barbara Anna Kowalczyk** [1,*] , **Monika Bieniasz** [2] **and Anna Kostecka-Gugała** [3]

1   Department of Ornamental Plants and Garden Art, Faculty of Biotechnology and Horticulture, University of Agriculture in Krakow, Al. Mickiewicza 21, 31-120 Krakow, Poland
2   Department of Horticulture, Faculty of Biotechnology and Horticulture, University of Agriculture in Krakow, Al. Mickiewicza 21, 31-120 Krakow, Poland; bieniasz.monika@urk.edu.pl
3   Department of Plant Biology and Biotechnology, Faculty of Biotechnology and Horticulture, University of Agriculture in Krakow, Al. Mickiewicza 21, 31-120 Krakow, Poland; anna.kostecka-gugala@urk.edu.pl
*   Correspondence: barbara.kowalczyk@urk.edu.pl

**Abstract:** Climate change is being felt in all vineyards around the world, opening up new perspectives for regions with a growing winemaking industry. In this study, 11 hybrid grapevines grown in cold climates were assessed in terms of flowering biology and pollination efficiency. The flowers were evaluated for the number of anthers and pollen grains in the flower; pollen viability and pollen grain size, the number of ovules in the ovary, and, consequently, the size and the weight of berries and the number of seeds in the berries were also analyzed. The flowers of *Vitis vinifera* L. usually have 5 stamens and 5 petals in their structure; this number for hybrid varieties ranged from 4 to 7, and in the case of the variety 'Seyval Blanc', it was 4 to 11 stamen and petals. Pollen grain size varied and ranged from 17.01 to 22.25 µm, while pollen grain pro-duction in flowers ranged from 5073 to 34,976 grain, which was calculated using a Bürker hemocytometer. The number of ovules in the ovary for the cultivars in question was highly variable, ranging from 3 to 7. One of the most important factors affecting flower pollination is stigma receptivity. Stigma receptivity appeared when the cap starts to fall off and disappeared at the browning of the cap. In connection with climatic changes, grapevine production is expanding to cool-climate countries. The aim of this study was to expand our knowledge about the flower morphology of 11 hybrid grapevine varieties most commonly cultivated in Poland. Knowledge of the flowering process can be important for improving yield and its quality.

**Keywords:** number grains; pollen viability; pollen size; ovule; pistils; stigma receptivity; seeds; grape cultivars; fruit quality

## 1. Introduction

The grapevine (*Vitis vinifera* L.) *Vitaceae* is the most widely cultivated and one of the most economically important fruit crops in the world. In Poland, it was not until the 1980s that the suitability of various grape cultivars for large-scale cultivation in colder climatic conditions began to be tested [1]. Currently, hybrid grape cultivars are mostly grown in Polish vineyards. Interspecific hybrid cultivars selected from the crosses of *V. vinifera* with species such as *V. rupestris*, *V. riparia*, *V. labrusca*, *V. berlandieri*, *V. lincencumii*, or *V. amurensis* are a small fraction of the global grapevine production, but are nonetheless very important locally. Increasing temperatures in Poland and the extension of the vegetation period contribute to the increase in grapevine acreage [2].

The quality of grapes depends on many factors including cultivar, terroir [3,4], management practices [5,6], and changing climate [7–10]. Ideal grapefruit composition, considering the sugar to acidity ratio, is obtained when grapes are produced in a temperate climate [10]. The last three decades have been warmer than any other since 1850 [11]. This forces growers to adapt their vineyards by selecting appropriate plant material and cultivation techniques.

Another method is shifting the cultivation area, which is also associated with a change in the profile of wines produced [11–14].

Acceptance of hybrid grape cultivars and their recognition by world oenologists will allow the cultivation of these grape cultivars to be improved [15]. There is a strong conservatism in viticulture in terms of recognizing only the *V. vinifera* cultivars. Hybrid cultivars possess a combination of characteristics that are superior to those of *V. vinifera* grapevines. Increasingly warmer summers are affecting the quality of white grapes in particular, due to a decrease in fruit acidity associated with malic acid degradation [16]. In colder regions, intense wines with a complex aromatic expression can be obtained using hybrid cultivars. Summer droughts, on the other hand, will favor the production of dark cultivars, for which water shortages at certain stages reduce the size of the berries and increase their polyphenol content [7,11,15,17–19]. The production of grapes with the quality parameters required by winemakers will largely depend on the environment and viticultural practices. There is a large seasonal variability in yields in a climate described as cool.

Fruit and seed production of most crops increases when cross-pollination occurs. Although self-pollination is a widespread biological process in grapevine cultivation, there are still some uncertainties regarding the consequences of cross-pollination on grapevine productivity and quality. The biology of grapevine fertilization becomes an important issue, especially when cover cropping is considered [20]. The yield and quality of grapes are markedly enhanced by cross-pollination although self-pollination ensures some level of berry set. Cross-pollination is also employed to breed a new fruitful genotype to sustain the food to feed the increasing world population in the face of climate change [21]. The flowering and pollination process itself is well understood in *Vitis vinifera*, while the cultivation of hybrid species still lacks comprehensive information on this topic. This knowledge would be of great cognitive importance. *V. vinifera* flowering and flower morphology of the vine have been extensively described by many authors [20–32]. Due to the complexity of inflorescence formation, the authors divided generative development into 22 stages designated as BBCH (Biologische Bundesanstalt, Bundessortenamt und Chemische Industrie) encoded with numbers from 0 to 50 based on the phenological development of the grapevine [25,33]. These stages coincide with the sum of active temperatures of each phase, e.g., in highbush blueberry to the onset of flowering 800 degrees [34] and apple trees to harvest around 2600 degrees [35]. The flowering process is known as anthesis and lasts about a week. The rate at which flowers develop depends primarily on environmental factors such as light and temperature, but also on internal factors such as hormonal changes. Low temperatures, which often occur in cold climates, can damage flower tissues and affect ovule development and pollen tube growth. The response to cold stress varies among genotypes [30,36,37]. The flower clusters of grapevines are quite inconspicuous. They are panicles (loose, irregularly branched flower clusters) with individual flowers, or blossoms, on the end of each branch. The flower consists of 5 sepals, 5 petals, 5 stamens, and a pistil ending in a stigma. The stigma of *Vitis vinifera* is of the solid type. It consists of a layer of the epidermis, parenchymatous tissue, and central conducting tissue. The epidermis consists of a single layer of tannin-rich cells [38]. The receptivity of a stigma is related to its structure. The stigma of *Actinidia deliciosa* flowers has a similar structure to that of *Vitis vinifera*, which has a surface covered with papillae; the stigma is covered with an abundant secretion when the flower is open. Papillae are mostly unicellular and contain numerous phenolic components. During the life of the flower, these papillae gradually lose their turgor, and from the opening of the flower (anthesis), they begin to burst their contents are released into the environment where the stigmas germinate. When the flower opens, the receptivity of the stigma is high and lasts for several days depending on the weather. For *Lonicera* spp., which also has a wet nevus type, it has been observed that receptivity decreases when the papillae rupture and the nevus tissue browns [39,40]. It was observed that hand-pollination *Vitis coignetiae* carried out at 0, 2, 4, and 6 days after anthesis showed that stigmas were most receptive two days after flower opening [32]. Pollen contains many nutrients that

are used by flower pollinating organisms. Bees require 10 amino acids such as arginine, histidine, lysine, tryptophan, phenylalanine, methionine, threonine, leucine, and valine for proper development. Among the plant species examined so far in this respect, among the species of fruit plants there is a large variation in the percentage of these substances, and thus in the attractiveness of pollen for pollinating insects [41]. Pollen viability and quantity are determining factors for successful pollination and fruit setting. To evaluate the biological value of pollen, pollen viability, and germination capacity should be taken into consideration. The germination capacity is subject to atmospheric factors including temperature during flowering and bud differentiation [34,40]. There are reports that the application of fungicides during flowering in Pinot Noir cultivar significantly reduced pollen germination [22]. The source of pollen has a significant effect on the percentage of fruit sets and their quality. The phenomenon of metaxenia is also observed, including a more intense berry color. Pollen sources obviously affected the berry detachment and skin rupture forces. The highest values regarding beery detachment and skin rupture forces were detected in 'Michele Palieri' × 'Italia' [21]. The petals are connected to each other by epidermal cells and form a cap (calyptra), whose function is to protect the flower's generative organs from environmental fluctuations during the early stages of bud burst. The cap falls off during blossoming and the individual flowers appear, and a phenomenon called capfall occurs. The released anthers burst and release pollen [25,29]. Recent studies have confirmed that the pollination process relies heavily on self-pollination and occurs before the capfall [29,42]. However, some cultivars of *V. vinifera* need cross-pollination, as protective mechanisms against self-pollination. One of these involves covering the stigma by the calyptra and lowering the anthers relative to the stigma [26,29]. In pollination, an important aspect is pollen fertility, i.e., its viability and germination. These traits will affect yield and are essential for fruit setting [43,44].

In connection with climatic changes, grapevine production is expanding to cool-climate countries. The aim of this study was to expand our knowledge about the flower morphology of 11 hybrid grapevine cultivars most commonly cultivated in Poland. Knowledge of the flowering process can be important for improving yield and its quality.

## 2. Materials and Methods

### 2.1. Biological Material

The experiment was set up in the South of Poland (50°08′29.4″ N 19°55′50.7″ E) in a temperate climate zone in 2018–2019. The research material consisted of 12-year-old vines of the following hybrid grapevine cultivars: Aurora, Bianca, Hibernal, Jutrzenka, Leon Millot, Marechal Foch, Muscat Odesskij, Regent, Rondo, Seyval Blanc, and Solaris. Table 1 details the origin of the hybrid grape cultivars discussed.

Vines were grown at 2.5 × 1 m in 24 rows of 100 of each variety spacing as a single Guyot pruning type, with permanent turf in between rows and herbicide strips 80 cm wide under vines. The vineyard lacked installed irrigation, and the average annual precipitation is sufficient in this region for vine growing (multi-year average of about 700 mm) (Figure 1B). Diseases and insects were controlled according to commercial guidelines.

**Table 1.** Origin of hybrid grape cultivars.

| Cultivar | Parents * | *V. Vinifera* (%) | *V. rupestris* (%) | *V. riparia* (%) | *V. labrusca* (%) | *V. berlandieri* (%) | *V. lincencumii* (%) | *V. amurensis* (%) | Flowering Date of Grape Cultivars ** |
|---|---|---|---|---|---|---|---|---|---|
| Aurora | Seibel 788 × Seibel 29 | 68.75 | 18.75 | 12.5 | | | | | medium-early |
| Bianca | Eger 2 × Bouvier | 78.09 | 14.58 | | 1.56 | 3.13 | 2.64 | | medium-late |
| Hibernal | Chancellor × Riesling | 82.04 | 14.06 | 1.95 | 1.95 | | | | late |
| Jutrzenka | SV 12-375 × Pinot Blanc | 78.01 | 14.58 | | 1.56 | 3.12 | 2.64 | | late |
| Leon Millot | MGt101 OP (V.rip × V. rup.) × Goldriesling | 50 | 25 | 25 | | | | | medium-late |
| Marechal Foch | MGt101-14 (V.rip × V. rup.) × Goldriesling | 50 | 25 | 25 | | | | | medium-late |
| Muscat Odesskij | Muskat Sinyj Rannij × CV 12-375 | 78.1 | 14.58 | | 1.56 | 3.12 | | | medium-early |
| Regent | Diana (Ga 30N-8-127) × Chambourcin | 80.06 | 14.32 | 0.98 | 1.76 | 1.56 | 1.38 | | medium-late |
| Rondo | Zarya Severa × Sait Laurent | 75 | | | | | | 25 | medium-early |
| Seyval Blanc | s. 4995 × s.4986 | 43.75 | 28.15 | 12.5 | 12.5 | 3.1 | | | late |
| Solaris | Merzling × Geisenheim 6493 | 73.4 | 7 | 3.1 | | 3.1 | 0.8 | 12.5 | medium-early |

* Prepared on the basis of data from the Julius Kühn Institut—Federal Research Centre for Cultivated Plants (VIVC). ** According to observations in the vineyard.

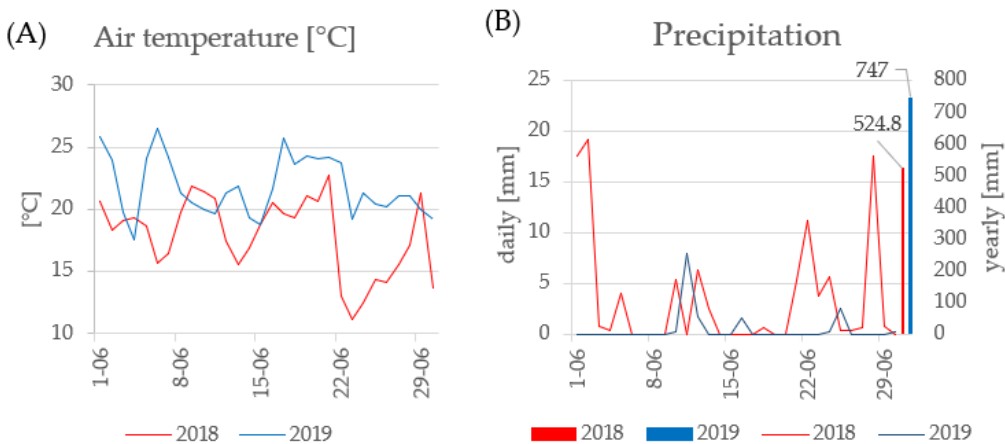

**Figure 1.** (**A**) Average daily temperatures recorded during the flowering period in 2018–2019, (**B**) Average daily precipitation recorded during the flowering period in 2018–2019.

Weather Data

Meteorological data from the 2018–2019 flowering period was collected from the iMetos go IMT meteorological station located in the vicinity of the vineyard.

### 2.2. Biometric Measurements and Flower Quality

2.2.1. The Number of Anthers

Flowers of 11 hybrid grapevine cultivars were collected in early June and analyzed for the number of anthers in the flower. A random sample of 100 fully developed flowers, from 50 plants, was taken for analysis (4 replicates of 25 each).

2.2.2. The Number of Ovules in the Ovary

In the next step, the number of ovules in the ovary was calculated. Measurements were carried out in four replicates, each replicate consisting of 25 ovaries. The ovaries were cut with a scalpel and the ovules were extracted using a micro-needle. Calculations were performed using a Carl Zeiss Discovery stereoscopic microscope.

2.2.3. Stigma Receptivity

The receptivity of the grapevine stigma was determined [45,46]. Flowers were taken from each cultivar and classified according to the following developmental stages: stage 1—closed flower (with calyptra); stage 2—flower with calyptra cut off but covered anthers; stage 3—flower without calyptra with anthers exposed; stage 4—flower with brown anthers. In the laboratory, a 3% $H_2O_2$ solution was applied to the stigma surface to assess receptivity and the stigma was observed under a Carl Zeiss Discovery 2.0 binocular magnifier. A receptive stigma actively releases oxygen in the form of gas bubbles, whereas a non-receptive stigma does not exhibit this ability. The percentage of receptive pistils at each flower stage was calculated (ten flowers were evaluated at each stage).

2.2.4. The Number of Pollen Grains in the Anther

The pollen yield of 11 selected cultivars was evaluated using a Bürker's hemocytometer by calculating the number of pollen grains in anthers and flowers. The sample consisted of 10 anthers randomly collected from 10 flowers of a given cultivar; the calculations were replicated four times. Observations were carried out using a Carl Zeiss Image M2 AXIO microscope at ten-fold magnification under a white light [47].

### 2.3. Quality and Quantity of Pollen Grains in the Flowers

2.3.1. Pollen Grains Size

Pollen grains size was measured using a light microscope. Pollen was selected from one hundred randomly selected flowers. The pollen mixture was spread onto slides, pollen size measurements were carried out in four replicates, one replicate consisted of one hundred pollen grains. The Axio vs. 40V software was used.

2.3.2. Pollen Grains Viability

Anthers were collected from 50 flowers for each cultivar and placed in Petri dishes and subsequently kept for 24 h at 25 °C for the anthers to open. The viability of pollen grains was assessed using Alexander's staining method [48], i.e., triple staining of pollen grains (malachite green, orange G, and acid fuchsin). Malachite green does not pass through the living cell membrane in viable pollen grains, and then the living pollen protoplast stains carmine, while the membrane of dead pollen is broken down and the pollen cytoplasm stains green.

According to Tello et al. [29], viability was divided into five groups determining pollen viability in percentage:

1. Very high viability > 90%;
2. High viability: 90–75%;
3. Medium viability: 75–50%;
4. Low viability: 50–25%;
5. Very low viability < 25%

### 2.4. Assessment of Pollination and Overgrowth of the Pollen Tube by the Pistil Neck

The number of germinating grains on the stigma and the number of pollen tubes at the stylopodium were calculated for the eleven cultivars. The experiment was performed in four replicates, each replicate consisting of twenty pistils with ovaries. The pistils were fixed in FAA (formalin:ethyl alcohol:acetic acid, 8:1:1) for 10 to 12 h. The pistils with ovules were subsequently macerated in 30% NaOH solution for two to three hours, and then the tissue was cleared with a 6% $H_2O_2$ solution. In the next step, after washing with water, the pistils with ovules were stained with aniline blue for three hours. In the final stage, the preparations were closed with glycerol, and observations were conducted using a Carl Zeiss Image M2 Axio fluorescence microscope [49] under ultraviolet light at a wavelength of about 356 nm. Under these conditions, callose fluoresces bright yellow-green and contrasts strongly with the bluish or grayish fluorescence of the stylar tissue. The pollen tubes are outlined by a callose lining and irregularly spaced callose plugs [50].

### 2.5. Evaluation of Fruit Quality

The fruit clusters were randomly selected for each cultivar and for each of the four replicates. From each cluster, four berries were collected from different levels of the cluster. The sample consisted of forty berries and four replicates. Biometric measurements of fruit such as berry weight, length, and width, number of seeds per berry, and berry spherical area were taken to assess fruit quality. The correlation of berry weight to seed number was calculated.

### 2.6. Statistical Analysis

The results were statistically analyzed using one-way analysis of variance. Tukey's post hoc test was used to assess the significance of differences between means, at the significance level of $\alpha = 0.05$. In addition, cluster analysis (Ward's method) was performed for biological parameters (number of pollen grains, anthers, ovules, and seeds). Statistical analysis was performed using the STATISTICA 13 software.

## 3. Results and Discussion

### 3.1. Biometric Measurements and Flower Quality

3.1.1. The Number of Anthers in Flowers of 11 Hybrid Grape Cultivars

When comparing the average number of anthers between the 11 hybrid grape cultivars and between seasons, it can be seen that the differences were small. The cultivar 'Hibernal' had the lowest number of anthers in both years of the experiment, and the cultivars 'Jutrzenka' and 'Marechal Foch' in 2019 (Figure 2A). The cultivars 'Rondo' and 'Solaris' had the highest number of anthers in the first year of the experiment, while the cultivars 'Solaris', 'Rondo', 'Aurora', 'Bianca', and 'Leon Millot' were in the group with the highest number of anthers in the second year. The average number of anthers (in absolute values) of the evaluated cultivars in both years of the experiment is comparable, only in the case of three cultivars 'Jutrzenka', 'Marechal Foch', and 'Seyval Blanc', it is slightly lower. As the number of anthers in flowers in the analyzed samples varied greatly, Figure 2B shows the percentage of flowers with 4, 5, 6, 7, 9, and 11 anthers. Most of the flowers of the discussed cultivars were characterized by a typical 5-fold flower structure (5 stamens and 5 petals), while exceptions to this rule were noted, including flowers with 4 to 11 anthers and petals, respectively (Figures 2B and 3A).

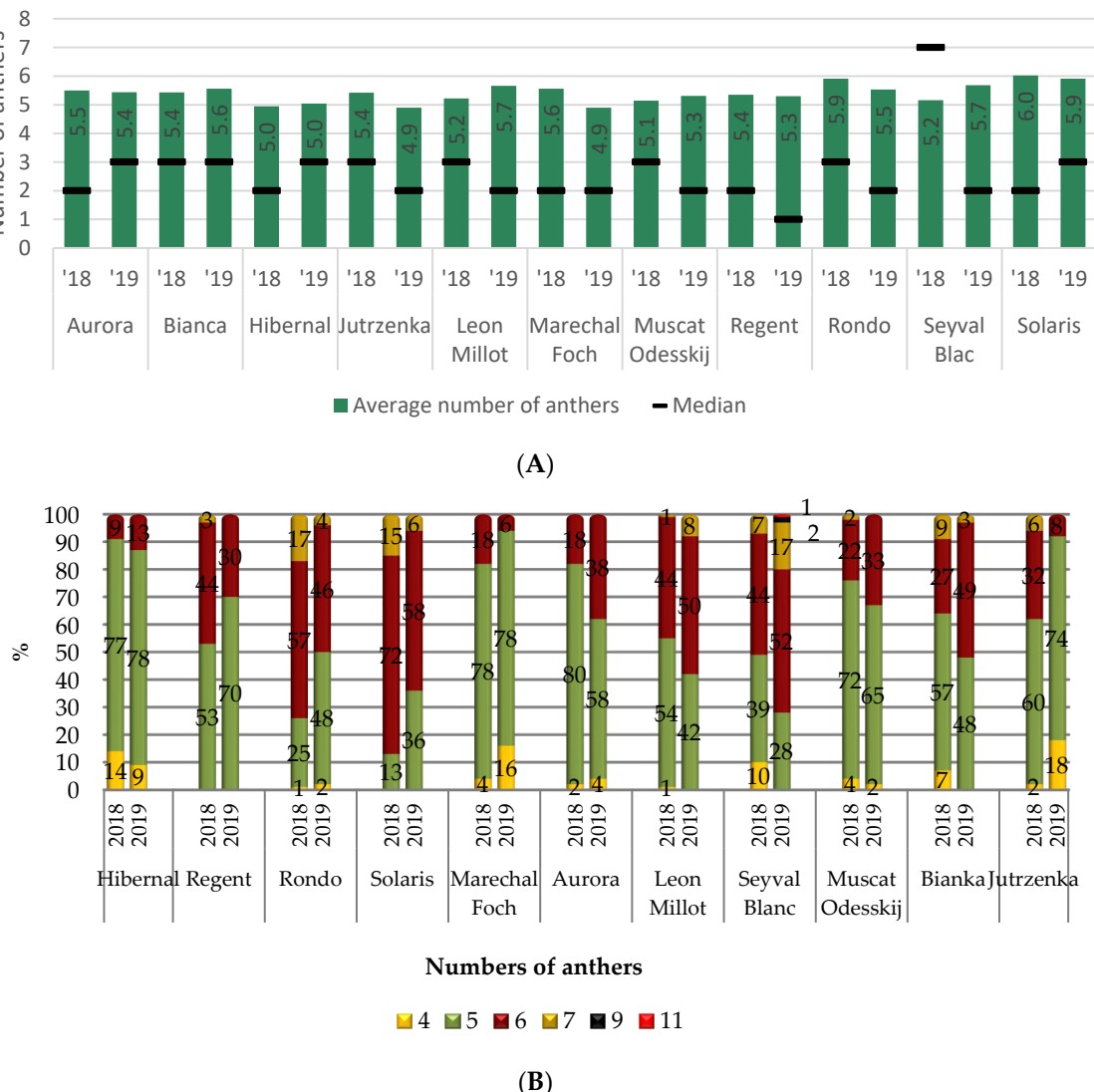

**Figure 2.** (**A**) Average number of anthers in flower of hybrid grapevine cultivars. (**B**) Variability in anther number in flowers of hybrid grapevine cultivars (%).

The number of anthers was always the same as the number of petals in the present study (Figure 3A) and in studies by other authors [29,51]. The group of cultivars that had the most flowers with 6 anthers included 'Solaris', 'Rondo', 'Leon Millot', and 'Seyval Blanc', recorded flowers containing 7 anthers in the same cultivars. In 'Seyval Blanc', flowers with 9 and 11 anthers were observed (Figure 3B). Kelen and Demirtas [52] analyzed 8 *Vitis vinera* cultivars grown in Turkey and also found differences in the number of anthers depending on the cultivar. The number of anthers for most of the analyzed cultivars was 5.0, only for two cultivars the average number of anthers varied between 5.6 and 6.0. Biasi and Conner [53] analyzed the flower structure of hermaphroditic muscadine cultivars (*Vitis rotundifolia* Michx.). These authors noted between 5 and 10 anthers per flower depending on the cultivar, with 6 to 8 anthers per flower being the most common. Padureanu and Patras [44] studied generative organs of two hybrid grapevines, 'Noah' and 'Otello', and reported that the number of anthers ranged from 4 to 6. Pierozzi and Moura [54] observed from 4 to 7 anthers in two mutants of the cultivar 'Niagara'.

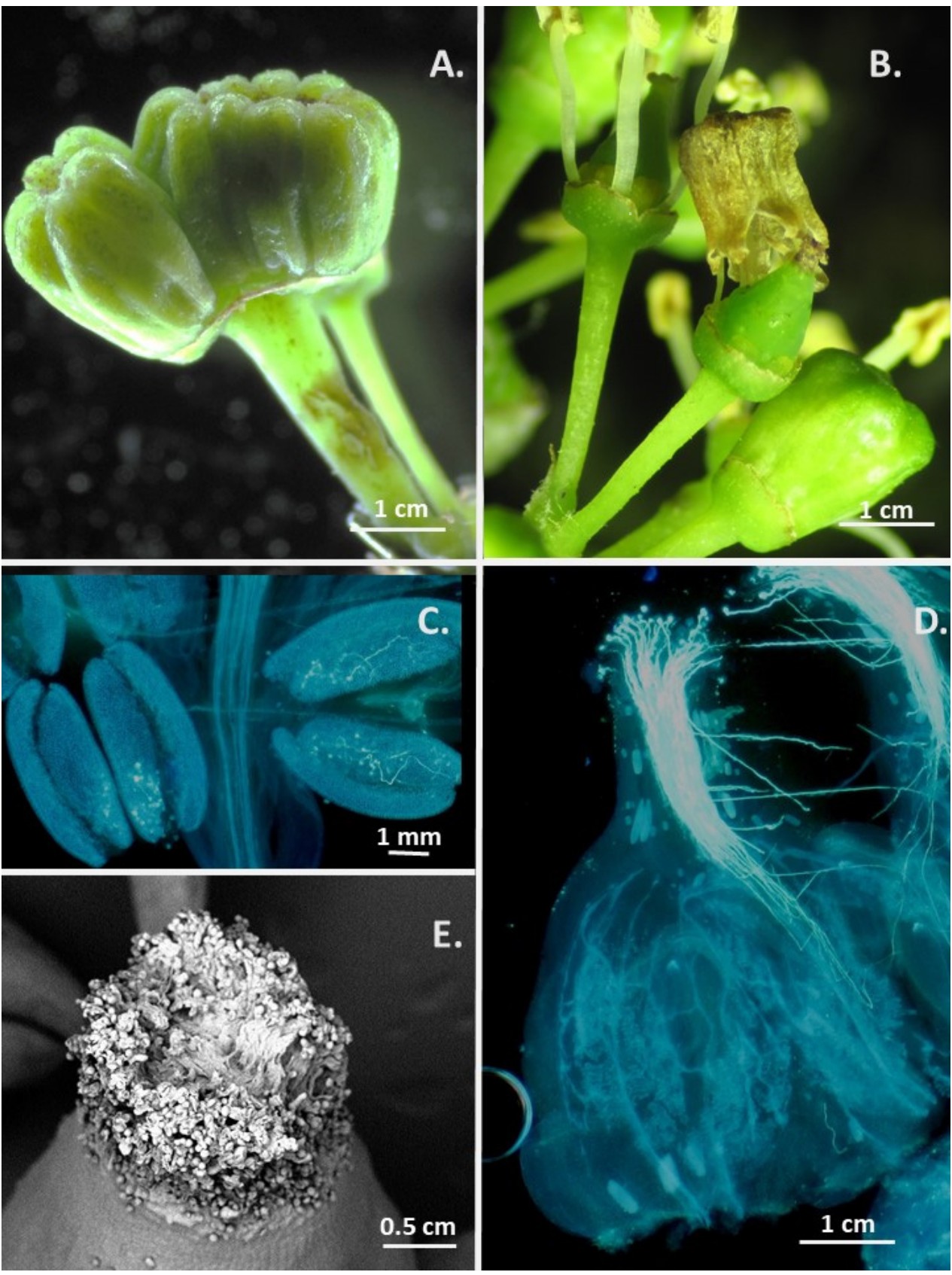

**Figure 3.** (**A**) Closed flower bud of a 'Seyval Blanc' cultivar with 11 anthers. (**B**) Capfall. (**C**) Pollen grains germinating inside closed anthers. (**D**) Pollen tubes growing through the pistil neck into the ovary. (**E**) Pollinated stigma of the 'Solaris' cultivar of pistil.

3.1.2. The Number of Ovules in the Ovary of 11 Grapevine Cultivars

In this study, the average number of ovules in the flowers of the discussed cultivars was calculated (Table 2). Statistically, the cultivar 'Solaris' had the highest number of seeds, and 'Jutrzenka' and 'Marechal Foch' the lowest. The ovary of the grapevine consists of two chambers, which usually contain two ovules [26]. In the present study, from 3 to 7 ovules were observed.

**Table 2.** Pollen viability and pollen tube outgrowth through the pistil neck.

| Cultivar | Pollen Viability% | | Average Number of Germinating Grains | Average Number of Pollen at the Base of the Pistil | Average Number of Ovules | |
|---|---|---|---|---|---|---|
| | 2018 | 2019 | | | | Min–Max |
| Aurora | 78.0 [cd] | 93.3 [d] | 40.8 [c] | 30.1 [d] | 4.6 [ab] | 4–6 |
| Bianca | 89.5 [d] | 74.0 [bcd] | 40.9 [c] | 24.2 [d] | 4.6 [ab] | 4–6 |
| Hibernal | 47.0 [b] | 50.7 [abc] | 0.8 [a] | 0.1 [a] | 4.6 [ab] | 4–6 |
| Jutrzenka | 63.5 [bc] | 78.0 [bcd] | 12.1 [b] | 2.1 [ab] | 4.3 [a] | 3–6 |
| Leon Millot | 59.5 [b] | 47.7 [abc] | 1.7 [a] | 1.7 [ab] | 5.2 [b] | 3–6 |
| Marechal Foch | 53.5 [b] | 45.7 [ab] | 40.5 [c] | 29.2 [d] | 4.2 [a] | 3–7 |
| Muscat Odesskij | 25.5 [a] | 78.0 [bcd] | 32.2 [bc] | 9.1 [bc] | 4.6 [ab] | 3–6 |
| Regent | 60.0 [b] | 91.7 [d] | 19.5 [b] | 6.7 [bc] | 4.8 [ab] | 3–6 |
| Rondo | 81.5 [d] | 86.0 [d] | 9.2 [b] | 0.3 [a] | 5.0 [ab] | 4–7 |
| Seyval Blac | 49.0 [b] | 37.0 [a] | 17.2 [b] | 1.3 [ab] | 5.2 [b] | 4–7 |
| Solaris | 84.0 [d] | 80.3 [cd] | 43.5 [c] | 21.7 [cd] | 6.1 [c] | 4–7 |

The values are given as means, followed by the letters [a–d] to indicate statistical significance. The values marked with the same letters in one column are not statistically different at $\alpha < 0.05$.

Cluster analysis (Figure 4) distinguished two groups of the studied cultivars that characterized quantitatively the generative organs. The analysis was based on the number of pollen grains in the flower, the number of ovules in the ovary, and the number of seeds in the berry. The group containing 'Rondo' and 'Solaris' cultivars, which were characterized by a high number of ovules, pollen grains per flower, and seeds, was clearly separated. The second group consisted of more combinations and two subgroups could be distinguished within it. The first subgroup included 'Hibernal', 'Jutrzenka', 'Bianca', 'Seyval Blanc', and 'Leon Millot' cultivars, and the second subgroup 'Muscat Odesskij', 'Regent', 'Aurora', and 'Marechal Foch' cultivars. In these groups, we observed similarity in terms of the average number of pollen grains in flowers, ovules, and seeds.

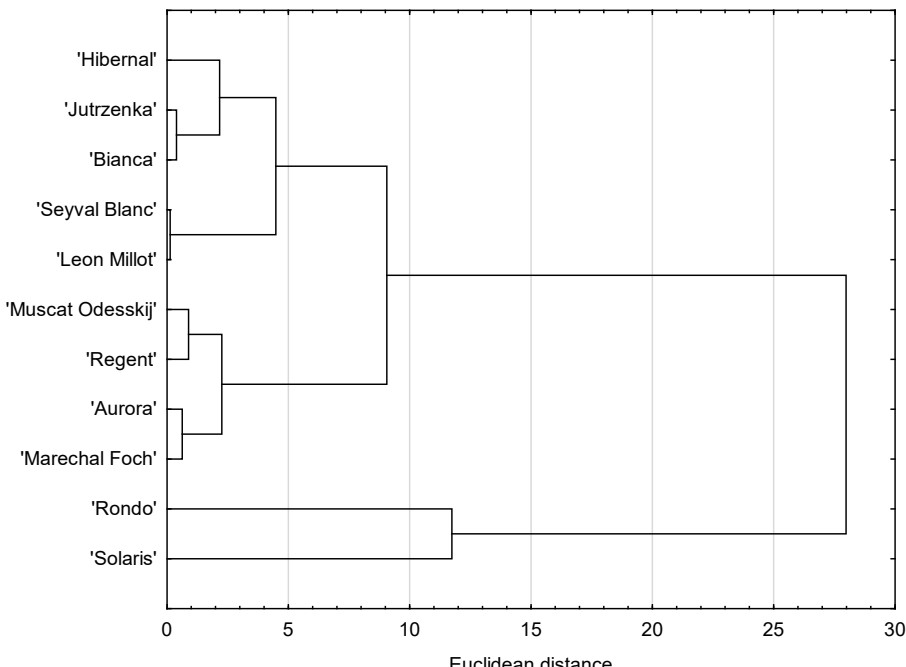

**Figure 4.** Cluster analysis characterizing the generative organs of the discussed cultivars in terms of the quantity of 11 hybrid grapevine cultivars.

### 3.1.3. Stigma Receptivity

Stigma receptivity, defined as the ability of the flower to accept pollen, changes during its life. Four stages (Figure 5) of flower development were distinguished for grapevine flowers. Table 3 shows that stigma receptivity varied depending on the stage of flower development. The receptivity of flower stigmas in the analyzed grapevine cultivars appears with the capfall (Figure 3B) and disappears when the stigma turned brown (Table 3, Figure 3E). Baby et al. [55] also found no significant differences in stigma receptivity between the three studied cultivars, i.e., 'Cabernet Sauvignon', 'Merlot', and 'Shiraz'.

**Table 3.** Mean percent flower pistil receptivity of 11 grapevine cultivars in relation to phenological phases.

| Cultivar | Immediately after Capfall | Full Flowering | Dried Anthers | Stigma Discoloured to Brown |
|---|---|---|---|---|
| Aurora | 83.5 ± 0.67 | 96.2 ± 0.12 | 35.4 ± 0.12 | 9.1 ± 0.06 |
| Bianca | 84.1 ± 0.06 | 96.2 ± 0.06 | 35.9 ± 0.15 | 8.6 ± 0.09 |
| Hibernal | 82.8 ± 0.12 | 96.4 ± 0.19 | 38.3 ± 0.09 | 8.8 ± 0.07 |
| Jutrzenka | 82.7 ± 0.12 | 97.0 ± 0.09 | 36.5 ± 0.25 | 8.5 ± 0.15 |
| Leon Millot | 82.8 ± 3.94 | 97.1 ± 0.20 | 35.4 ± 1.82 | 8.7 ± 0.17 |
| Marechal Foch | 82.0 ± 0.06 | 96.3 ± 0.27 | 36.9 ± 0.06 | 8.5 ± 0.18 |
| Muscat Odesskij | 84.5 ± 0.12 | 97.2 ± 0.49 | 35.9 ± 0.29 | 8.6 ± 0.34 |
| Regent | 82.4 ± 0.12 | 97.1 ± 0.06 | 36.9 ± 0.42 | 8.4 ± 0.12 |
| Rondo | 83.4 ± 0.12 | 96.7 ± 0.06 | 37.8 ± 0.09 | 8.8 ± 0.20 |
| Seyval Blanc | 83.4 ± 0.12 | 96.9 ± 0.12 | 36.1 ± 0.18 | 8.7 ± 0.12 |
| Solaris | 83.8 ± 4.30 | 96.5 ± 0.25 | 35.7 ± 0.21 | 9.0 ± 0.03 |

The values are given as means ± standard deviations. Not statistically different at $\alpha < 0.05$.

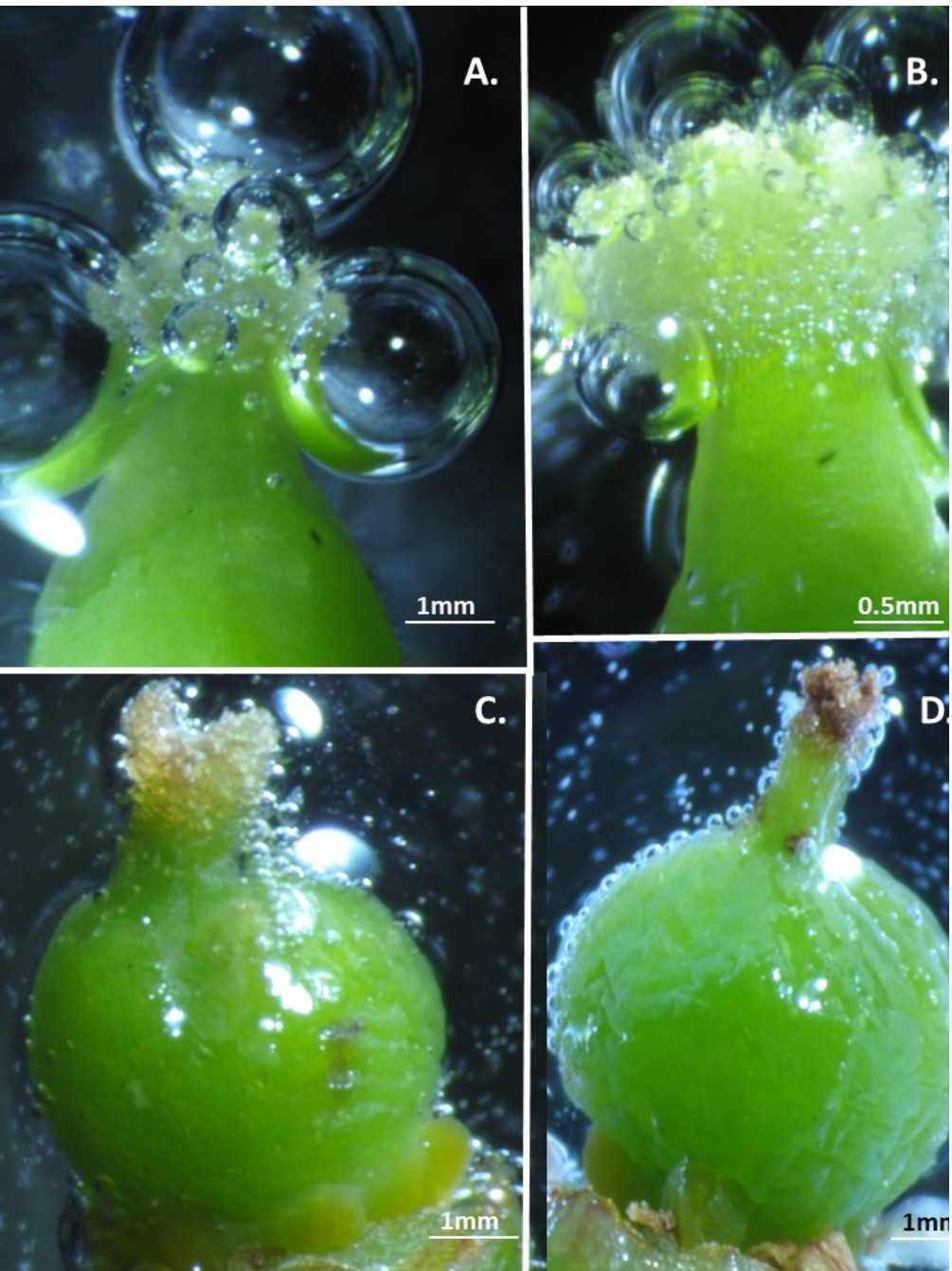

**Figure 5.** Stigma receptivity. (**A**)—Immediately after capfall, (**B**)—Full flowering, (**C**)—Dried anthers, (**D**)—Stigma discolored to brown.

### 3.1.4. The Number of Pollen in the Anther

The number of pollens in the anther varies greatly depending on the species and cultivar as well as weather conditions during microsporogenesis [40,47,49]. The group of cultivars with the lowest number of pollen grains in the anther in the first year of the experiment included 'Hibernal', 'Seyval Blanc', 'Leon Millot', and 'Regent' cultivars (1162.5–1981.25), while the lowest number of pollen grains in the second study year was observed for 'Hibernal', 'Dawn', and 'Bianca' cultivars (912.34–1657.81) (Table 4). The cultivar 'Solaris' in both years of the experiment was characterized by the highest number of pollen grains per anther and flower (Table 4). A high number of pollen grains in the anther and in the flower in the first year of the experiment was characterized by the cultivar

'Rondo', while in the second year 'Rondo' and 'Regent'. Kelen and Demirtas [34] recorded from 2906 to 9000 pollen grains per flower. Comparing the results obtained by the authors of the present study with the results of Kelen and Demirtas [52], it could be seen that the values were significantly higher. The number of pollen grains per anther in muscadin cultivars in the study of Biasi and Conner [53] ranged from 2539 to 5776.8, while per flower it was from 16,380 to 39,860. These values were more similar to the authors' results.

**Table 4.** Biometric measurements of flowers of 11 hybrid grape cultivars.

| Cultivar | Average Pollen Size Diameter [µm] | | Average Number of Grains per Anther | | Average Number of Grains per Flower | |
|---|---|---|---|---|---|---|
| | 2018 | 2019 | 2018 | 2019 | 2018 | 2019 |
| Aurora | 21.3 [ef] | 21.3 [ef] | 2618.7 [de] | 2631.2 [d] | 14403.1 [e] | 14314.0 [cd] |
| Bianca | 22.2 [f] | 22.2 [f] | 2368.7 [cde] | 912.3 [a] | 12909.7 [cde] | 5072.6 [a] |
| Hibernal | 19.4 [cd] | 19.6 [cd] | 1162.5 [a] | 1657.8 [c] | 5754.4 [a] | 8355.4 [a] |
| Jutrzenka | 17.6 [ab] | 17.8 [ab] | 2212.5 [cd] | 1281.2 [b] | 11991.7 [cde] | 6278.1 [a] |
| Leon Millot | 21.6 [f] | 21.8 [f] | 1862.5 [bc] | 2300.6 [d] | 9722.2 [bc] | 13021.5 [c] |
| Marechal Foch | 19.5 [cd] | 19.5 [cd] | 2337.5 [cde] | 3025.0 [e] | 12996.5 [de] | 14852.7 [cd] |
| Muscat Odesskij | 20.5 [e] | 20.5 [de] | 2863.5 [e] | 3296.9 [e] | 14713.2 [e] | 17506.4 [e] |
| Regent | 19.0 [c] | 19.2 [c] | 1981.2 [bc] | 3914.1 [f] | 10599.7 [bcd] | 20744.5 [g] |
| Rondo | 18.8 [bc] | 19.0 [c] | 4020.3 [f] | 4153.1 [f] | 23760.0 [f] | 22925.2 [g] |
| Seyval Blac | 17.0 [a] | 17.0 [a] | 1537.5 [ab] | 2579.7 [d] | 7675.5 [ab] | 15271.7 [d] |
| Solaris | 18.5 [c] | 18.6 [bc] | 5801.6 [g] | 6157.8 [g] | 34925.4 [g] | 34976.4 [f] |

The values are given as means, followed by the letters [a–g] to indicate statistical significance. The values marked with the same letters in one column are not statistically different at α < 0.05.

### 3.2. Quality and Quantity of Pollen Grains in the Flowers

#### 3.2.1. Pollen Grains Size

The group of cultivars with pollen size above 20 µm included 'Aurora', 'Bianca', 'Leon Millot', and Muskat Odesskij cultivars. The smallest pollen grains were recorded for the cultivars 'Jutrzenka', 'Seyval Blanc', and 'Solaris', the other cultivars did not differ statistically (Table 4). Vasconcelos et al. [29] reported that grapevine pollen dimensions ranged from 25 to 30 µm in length and 12 to 15 µm in width. The same authors indicated that pollen fertility was determined by its shape, fertile pollen was barrel-shaped, while non-fertile pollen had an oblong shape. Similar observations were made by Baby et al. [55], who described spherical pollen as hydrated, capable of fertilization, while oblong pollen was considered dehydrated, incapable of fertilization. Pierozzi and Moura [54] recorded pollen with mean sizes of 20.67 and 26.43 µm for two mutants of the dessert cultivar 'Niagara'. İşçi [56,57] analyzed pollen of many *V. vinifera* cultivars and its size varied greatly between them (16.26–29.91 µm) [56]. The pollen of *V. vinifera* cultivars described by the author was much smaller compared to pollen sizes in the current experiment.

#### 3.2.2. Pollen Viability

Effective pollination and generation of well-formed seeds largely depend on pollen quality [52,55,58]. In the current experiment, the cultivar 'Muscat Odesskij' was characterized by low pollen viability in 2018, while 'Bianka', 'Aurora', 'Rondo', and 'Solaris' cultivars were included in the group of high pollen viability (Table 3). In 2019, pollen viability of the study cultivars was altered. In that season, the cultivar 'Seyval Blanc' belonged to the group with low pollen viability, while 'Regent' and 'Aurora' cultivars were included in to the group with very high pollen viability. Many authors have reported differences in pollen viability in *Vitis vinifera* [52,55,57,59,60]. Baby et al. [55] analyzed pollen viability in

three *Vitis vinifera* cultivars: 'Merlot', 'Cabernet Sauvignon', and 'Shiraz'. They showed a high proportion of low viability pollen in 'Merlot' and 'Cabernet Sauvignon' cultivars. Defouquette et al. [61] found higher pollen viability in muscadine and hermaphroditic cultivars (*Vitis rotundifolia*) compared to the present study, ranging from 69.8% to 99.8%, which was similar to the results of Biasi and Conner [53].

### 3.3. Assessment of Pollination and Overgrowth of the Pollen Tube by the Pistil Neck

In the current experiment, the number of germinating pollen grains on the stigma and the number of pollen tubes at the base of the pistil were calculated (Table 2, Figure 3E). Statistically, the highest number was observed for 'Solaris', 'Marechal Foch', 'Aurora', 'Bianca', and 'Muscat Odesskij' cultivars. The lowest number of pollen grains germinating on the stigma was recorded for 'Hibernal' and 'Leon Millot' cultivars. Ebadi et al. [62] also studied the growth of the pollen tube in the cultivars 'Chardonay' and 'Shiraz'. These authors observed that low temperature during flowering weakened pollen germination in the tested cultivars and was a cultivar-specific feature expressed by quantitative differences in the disturbance of pollen functions. In the present study, flowering in 2018 began on June 8, and in 2019 on June 10. It lasted, depending on the cultivar, from 7 to 14 days. In 2018, the lowest average daily temperature was recorded on June 17 (18.7 °C), while in 2019, on June 13 (15.56 °C) (Figure 1A). The rainfall in both years was at a similar level and did not cause any changes in flowering (Figure 1B). According to some authors [29,63], temperatures below 15 °C and rainfall delay flowering.

In our study, the lowest number of pollen tubes at the base of the pistil was recorded for the 'Hibernal', Seyval,' 'Leon Millot', 'Rondo', and 'Jutrzenka' cultivars (Table 2, Figure 3D). The highest number in turn was observed for 'Aurora', 'Bianca', and 'Solaris' (from 21 to 30%). Ebadi et al. [62] reported that 20% of ovules were penetrated by pollen tubes and fertilized in the cultivar 'Chardonay'. Padureanu and Patras [44], investigating the fertility of two hybrid cultivars, found that the cultivar 'Otello', which contained genes from *Vitis vinifera*, was characterized by 2 times higher pollen yield compared to the cultivar 'Noego' derived from *Vitis labrusca* and *Vitis riparia*. In the present study, all the cultivars studied had some share of *Vitis vinifera* (Table 1), and regardless of their genotype, were characterized by the variability in pollen germination and the number of pollen tubes at the stylopodium relative to the cultivar.

The authors observed pollen germination in closed anthers in 10% of flowers of all cultivars (Figure 3C). There are several theories about how grapevines are pollinated, ranging from cleistogamy, i.e., self-pollination, insect pollination, to wind pollination [29]. Staud et al. [64] performed observations of *Vitis vinifera* flower opening in 'Müller-Thurgau' and 'Pinot Noir' cultivars. They noticed that about 25–35% of the pollen showed pollen tube growth as early as in the capfall phase (Figure 3B). Similar observations were made by Defouquette et al. [61]. They reported that self-pollination in *Vitis roundifolia* Michx cultivars was approximately 22%. Maghradze [59] recorded self-pollination in Georgian cultivars at a level from 16.9% to 61.9%, and this trait was cultivar-specific. Depending on the cultivar, the authors indicated that self-pollination could have had a positive or negative effect on berry size, or the number of seeds produced; however, they added that apart from self-pollination, cross-pollination was also necessary for good fruiting. [42,59,61]. Pollination is usually positively correlated with the number of seeds per fruit [42,65] and it was also confirmed in our study.

### 3.4. Evaluation of Fruit Quality

Parameters such as the average berry weight, length, and width, along with spherical area and average number of seeds were considered in fruit quality evaluation. The correlation between berry weight and seed number in berry was also calculated (Table 5).

**Table 5.** Fruit quality 11 hybrid grape cultivars.

| Cultivar | Average Berry Weight | Average Berry Width | Average Berry Length | Globular Surface (cm²) | Average Number of Seeds in Berry | Correlation (between Berry Weight and Number of Seeds) |
|---|---|---|---|---|---|---|
| Aurora | 1.9 de | 14.0 ab | 14.6 b | 2.0 bcd | 2.6 b | 0.31 |
| Bianca | 2.0 de | 14.4 b | 15.3 b | 2.2 e | 2.4 b | 0.86 |
| Hibernal | 2.0 de | 13.9 ab | 14.8 b | 2.1 cde | 2.4 b | 0.65 |
| Jutrzenka | 1.9 cd | 13.5 ab | 14.3 b | 1.9 bcd | 2.2 ab | 0.47 |
| Leon Millot | 1.7 b | 13.0 ab | 14.1 b | 1.8 ab | 2.1 ab | 0.78 |
| Marechal Foch | 2.0 e | 13.9 ab | 15.4 b | 2.1 e | 2.4 b | 0.81 |
| Muscat Odesskij | 1.6 ab | 13.2 ab | 14.4 b | 1.9 abc | 2.3 b | 0.77 |
| Regent | 1.6 a | 12.8 a | 14.1 b | 1.8 ab | 1.6 a | 0.87 |
| Rondo | 1.9 d | 13.8 ab | 14.9 b | 2.1 cde | 2.3 b | 0.63 |
| Seyval Blanc | 1.8 bc | 13.4 ab | 15.0 b | 2.2 e | 2.1 ab | 0.79 |
| Solaris | 1.9 de | 13.9 ab | 14.5 b | 2.0 cd | 3.6 c | 0.91 |

The values are given as means, followed by the letters [a–e] to indicate statistical significance. The values marked with the same letters in one column are not statistically different at $\alpha < 0.05$.

The fertilized ovule develops into a seed [66]. The size of the berries and their shape in individual cultivars is genetically controlled; the weight of grape berries also depends on the number of seeds [42], i.e., berries without seeds will be much smaller than berries containing them. The same is true for other species such as apples [67]. This relationship in the current study could is observed particularly for the cultivar 'Regent', as its berries had the lowest weight, the smallest size, the smallest number of seeds (Table 5) and the highest percentage of berries with a single seed (Figure 6). Moreover, parthenocarpic berries were also found in this cultivar (Figure 6). Spena and Rotino [68] have argued that parthenocarpy in grapevines and other species is a way of producing fruit under conditions not conducive to pollination and/or fertilization. In Central European climatic conditions, parthenocarpic fruits very often develop from fruits damaged by spring frosts [67].

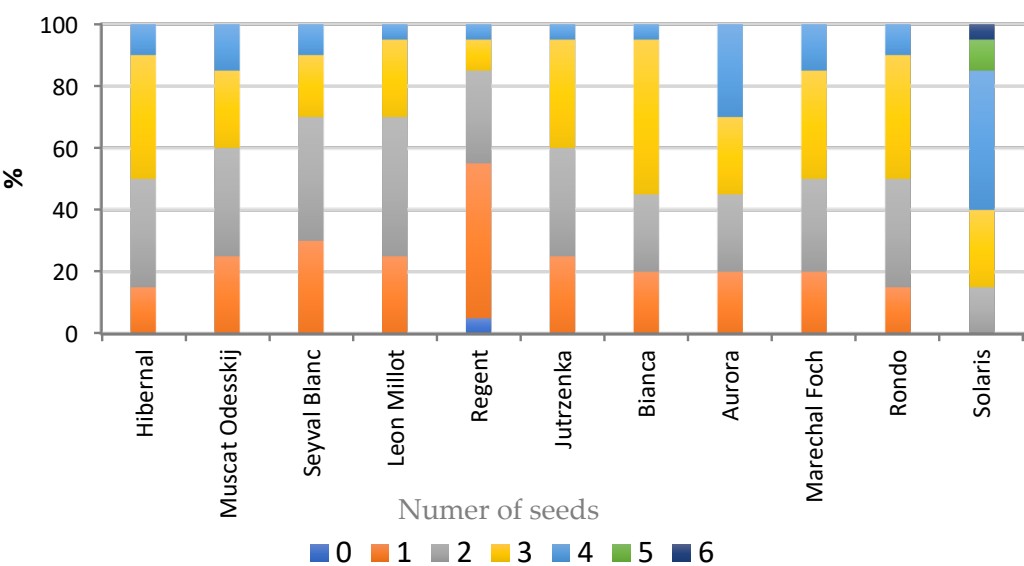

**Figure 6.** Percentage of number of seeds per berry in 11 hybrid grape cultivars.

The cultivar 'Solaris' was clearly deviating from Sabir's observations [42], as its berries did not increase in weight or size despite a significantly higher number of seeds (Table 5).

This cultivar had the highest percentage of berries with 4 seeds and a small percentage of berries with 5 and 6 seeds (Figure 4). Maghradze [59] found in his work that there was a possibility of producing more seeds in grapevine berries than usually observed (i.e., 0, 1, 2, 3, 4), due to aberrations during micro- and macrosporogenesis, embryogenesis, and genotype variation. The correlation between berry weight and number of seeds was the highest for this cultivar (Table 5). Lebon et al. [25] analyzed inflorescence and flower development of the cultivars 'Gewurztraminer' and 'Pinot Noir' under different conditions. The number of seeds per berry averaged from 1.4 to 1.8. In the present experiment, for most cultivars, the number of seeds was higher than Lebon et al. [25] and ranged from 1.6 to 2.45.

Cultivars with the highest spherical area in the present experiment included 'Seyval Blanc', 'Bianca', and 'Marechal Foch', while 'Regent' and 'Leon Millot' were characterized by the lowest (Table 5).

## 4. Conclusions

Hybrid cultivars differ from *Vitis vinifera* in terms of flower structure. We recorded anomalies in the number of anthers ranging from 4 to 7 for most of the described cultivars and up to 11 anthers in the cultivar 'Seyval Blanc'. The number of pollen grains in a flower varied greatly and depended on the cultivar and growing season. The cultivars 'Rondo' and 'Solaris' were characterized by the highest values of flower structure and quality (number of pollens in flowers, pollen viability, number of ovules in the ovary, number of set seeds). Stigma receptivity appered with the capfall and disappeared with its browning. The cultivar 'Regent' had the ability to form a small percentage of parthenocarpic fruits. Berry weight is not correlated with the number of seeds set. The existence of *Vitis vinifera* cultivars has been described in many publications, while hybrid cultivars require better knowledge, evaluation of the source of pollen on fruit quality (metaxenia), and checking fruit quality after pollination with their own pollen (kleistogamy). These issues require further research.

**Author Contributions:** Conceptualization, M.B. and B.A.K.; methodology, B.A.K. and M.B.; software, B.A.K.; validation, M.B.; formal analysis, B.A.K.; investigation, B.A.K. and M.B.; resources, B.A.K.; data curation, B.A.K.; writing—original draft preparation, B.A.K.; writing—review and editing, B.A.K., M.B. and A.K.-G.; visualization, B.A.K.; supervision, M.B.; project administration, B.A.K.; funding acquisition, M.B. and A.K.-G. All authors have read and agreed to the published version of the manuscript.

**Funding:** This research was supported by the Ministry of Education and Science of Poland as a part of a research subsidy to the University of Agriculture in Krakow.

**Institutional Review Board Statement:** Not applicable.

**Informed Consent Statement:** Not applicable.

**Data Availability Statement:** Not applicable.

**Conflicts of Interest:** The authors declare no conflict of interest. The funders had no role in the design of the study; in the collection, analyses, or interpretation of data; in the writing of the manuscript; or in the decision to publish the results.

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
