# Peer review of "Flowering Biology of Selected Hybrid Grape Cultivars under Temperate Climate Conditions"

_agriculture, doi:10.3390/agriculture12050655_

Round 1

Reviewer 1 Report

Please see reviewed file.

Author Response

The authors thank very much for valuable comments, which certainly contributed to the transparency of this work.

King regards, 

Barbara Kowalczyk

Reviewer 2 Report

Please find my comments and suggestion within the attached pdf file.

Author Response

(The authors gave the same response as above.)

Reviewer 3 Report

The manuscript is written with clear understanding of the project addressed. However, there are major concerns that need to be addressed to enhance the quality of the manuscript. My specific comments are as follows:

Abstract:

Elaborate more on the methods used in this study.

Keywords:

Delete number of pollen grains, size of pollen grains. Change to pollen grains.

Delete number of ovules, number of pistils, seeds numbers.

Add grape cultivars.

Introduction:

Page1Line31: “In Poland, it was not until the 1980s..” Add citation

Add the objectives of this study at the end of introduction.

Add literatures/other studies conducted on different grape cultivars related to the flowering process

P2L60: “V. vinifera flowering and flower morphology of the vine have been extensively described by many authors [13–18].” Elaborate on these studies.

Materials and Methods:

Table 1 should be cited in Section 2.1. What are the percent of grapevine varieties represented for?

Simplify the title for Section 2.2.1 and 2.2.2

Section 2.3.1 should be incorporated under Section 2.2. Please revise accordingly.

P6L168: “The sample consisted of forty berries.” Is this for each variety? Please explain in details regarding how many samples were used for overall process

Results:

Figure 3 is very confusing. Please revise the legend, title of figure, etc. I don’t understand how the legend is related to the calculation of flower anthers

P6L190: Where is Fig. 3.2a?

P7L197: “The authors recorded flowers with 7 anthers in the same varieties.” Which authors?

Table 3: “Not statistically different at α < 0.05” What does it mean by this statement?

P9L229: “They were similar in terms of the average number of pollen grains in flowers, ovules and seeds.” Which varieties are similar?

Section 3.2.1. The flow of discussion is a bit confusing. To make the readers easy to understand, discuss the number of grains per anther first, then followed by the discussion of number of grains per flower.

P11L277: “Ä°ÅžÇÄ° [39,40] analyzed pollen of many V. vinifera varieties and its size varied greatly between them (16.26-29.91 μm) [39].” Please revise this citation.

P13L330: “Defouquette et al. [44]. They reported that self-pollination in Vitis roundifolia Michx varieties was approx. 22%.” Delete they

P13L347: “..and the highest percentage of berries with a single seed (Fig.4). Moreover, parthenocarpic berries were also found in this variety (Fig.4).” Is it referring to the correct figure? Figure 4 shows the cluster analysis.

Where are the discussion for Figure 5 and 6? Please elaborate on these.

Conclusions:

Please revise and combine in one single paragraph. add on recommendation for future studies.

General comments:

Please check the reference styles and grammar of the manuscript.

Reviewer 4 Report

The authors presented that the influence of flowering biology in hybrid grape cultivars under temperate climate conditions. Generally, this manuscript is novel. After I reviewed this paper, I suggest a minor revision before acception.

minor comment:

Introduction is not well written, may need to be revised.

Is there any change of expression level of flowering-related genes?

Round 2

Reviewer 2 Report

Authors, I appreciate the time and effort you put amend the manuscript. I still believe it is necessary to consider using a robust data analysis tool for your data, as I have mentioned in the previous version. 
I can't read table 2; it appears untraceable to see. 
